# Staphylococcal Resistance Patterns, *blaZ* and SCC*mec* Cassette Genes in the Nasopharyngeal Microbiota of Pregnant Women

**DOI:** 10.3390/ijms24097980

**Published:** 2023-04-28

**Authors:** Sylwia Andrzejczuk, Monika Cygan, Dominik Dłuski, Dagmara Stępień-Pyśniak, Urszula Kosikowska

**Affiliations:** 1Department of Pharmaceutical Microbiology, Medical University of Lublin, W. Chodźki Str. 1, 20-093 Lublin, Poland; 2Student Research Group at the Department of Pharmaceutical Microbiology, Medical University of Lublin, W. Chodźki Str. 1, 20-093 Lublin, Poland; 3Department of Obstetrics and Perinatology, Medical University of Lublin, Jaczewskiego Str. 8, 20-090 Lublin, Poland; 4Department of Veterinary Prevention and Avian Diseases, Faculty of Veterinary Medicine, University of Life Sciences in Lublin, 20-950 Lublin, Poland

**Keywords:** nasopharyngeal microbiota, pregnancy, pregnant women, staphylococci, *blaZ* gene, SSC*mec* cassette typing

## Abstract

Antimicrobial resistance in *Staphylococcus* spp. colonising the nasopharynx can create risk factors of therapeutic treatment failure or prophylaxis in pregnant women. Resistance is mostly encoded on plasmids (e.g., *blaZ* gene for penicillinase synthesis) or chromosomes (e.g., *mecA* and *mecC* for methicillin resistance). The *mecA* gene is part of the chromosomal *mec* gene cassette (SCC*mec*), which is also located on the plasmid. The disc diffusion method for the selected drugs (beta-lactams, fluoroquinolones, streptogramins, aminoglicosides, macrolides, oxasolidinones, tetracyclines and other groups) was used. PCR for *blaZ*, *mecA* and *mecC* genes and SCC*mec* cassette detection and typing were performed. *S. aureus* (54.4%) and *S. epidermidis* (27.9%) were the most prevalent and showed the highest diversity of resistance profiles. The *blaZ, mecA* and *mecC* genes were reported in 95.6%, 20.6% and 1.5% of isolates, respectively. The highest resistance was found to beta-lactams, commonly used during pregnancy. Resistance to a variety of antimicrobials, including benzylpenicillin resistance in *blaZ*-positive isolates, and the existence of a very high diversity of SCC*mec* cassette structures in all staphylococci selected from the nasopharyngeal microbiota of pregnant women were observed for the first time. Knowledge of the prevalence of antimicrobial-resistant staphylococci in the nasopharynx of pregnant women may be important for the appropriate treatment or prophylaxis of this group of patients.

## 1. Introduction

During pregnancy, antibiotics should only be used for legitimate indications where untreated bacterial infection can adversely affect maternal health and fetal development [1]. The main indications for initiating antibiotic therapy in a pregnant woman are urinary tract infections, including asymptomatic bacteriuria; pyelonephritis; bacterial vaginosis; bacterial respiratory tract infections (including pneumonia); gonorrhoea; rheumatoid arthritis; syphilis; toxoplasmosis; premature rupture of the membranes of the fetus; group B streptococci (prophylaxis during childbirth); and other serious conditions (including endocarditis) [1,2].

Antibiotics are the most commonly prescribed class of drugs during pregnancy and lactation (80% of all prescribed drugs) [1]. It is estimated that more than 40% of women took an antibiotic during pregnancy as part of perinatal prophylaxis for early neonatal sepsis due to *Streptococcus agalactiae* (GBS) or for prophylaxis of surgical site infection after a caesarean section [2]. When considering other antimicrobial therapies in pregnancy, such as the treatment of asymptomatic bacteriuria or bacterial vaginosis (BV), and the overuse of antibiotics to treat respiratory and genital tract infections, it is reasonable to assume that the vast majority of fetuses are now exposed to antibiotics before birth [1,2,3]. The following antibiotics may be used during pregnancy: penicillins (amoxicillin and ampicillin), cephalosporins (cefazolin, cephalexin, ceftriaxone and cefuroxime), macrolides (erythromycin) and cotrimoxazole (contraindicated in the first trimester of pregnancy) [2,4,5]. According to the Polish recommendations, all women undergoing an elective or emergency caesarean section should receive antibiotic prophylaxis, and a single dose of first-generation cephalosporins is the drug of choice [2]. If the patient is allergic to penicillin, clindamycin or erythromycin may be used [2,4,5]. Further research is needed on antibiotic prophylaxis in operative vaginal delivery [2]. The World Health Organisation (WHO) recommends the use of antibiotic prophylaxis despite the lack of scientific evidence of its effectiveness and emphasises the need for appropriate research, particularly in caesarean sections and other manipulative intrauterine procedures [1]. To reduce the risk of GBS infection, perinatal prophylaxis is given with penicillin or ampicillin and, in cases of penicillin allergy, with cefazolin (without the risk of anaphylaxis), clindamycin, erythromycin or vancomycin [1,4,5].

Staphylococci are an important component of healthy human microbiota. They can occur within the skin glands and on the skin as major colonisers, as well as on the surface of the mucous membranes of the respiratory and urogenital systems [6,7,8,9]. In the study by Fujiwara et al. [10], determining the composition of the oral microbiota in both women at different pregnancy periods (early, 7–16 weeks gestation; middle, 17–28 weeks; and late, 29–39 weeks) and healthy non-pregnant women, the genus *Streptococcus* was found to be the most abundant, followed by *Staphylococcus*. Staphylococcal infections are classified as invasive, with a severe course, especially in immunodeficient individuals. *Staphylococcus aureus* has long been known to cause a wide range of infections, from mild to fatal [7]. Coagulase-negative *Staphylococcus epidermidis* (CoNSE) and other coagulase-negative staphylococci (non-SE CoNS) in recent years have become one of the most important causes of nosocomial infections [8,9,11,12,13,14,15,16]. It is also known that in 1.5-month-old infants, *Staphylococcus* spp. is predominant in the nose and nasopharynx [17].

Over the last few years, their resistance to antimicrobials has increased significantly. Many of them have become known as multidrug-resistant strains [11]. The resistance is most often encoded on plasmids, as in the case of the *blaZ* gene, which determines the synthesis of penicillinases [18,19,20]. The second type of resistance is encoded on the bacterial chromosome, as in the case of the *mecA* and *mecC* genes responsible for methicillin resistance [8,9,16,21,22]. The *mecA* gene is part of the staphylococcal chromosomal *mec* gene cassette (SCC*mec*), located within the plasmid [23,24]. The origin of the SCC*mec* cassette is not precisely known; there are indications that *S. epidermidis* may be its reservoir of resistance genes, mainly for the more pathogenic *S. aureus* [12,14]. SCC*mec* cassettes differ from one another. The cassette consists of the *mecA* operon, the *ccr* gene complex, and three joining regions. It is bounded at both of its ends by repeptide structures. These elements are the constants of the basic backbone of such a gene cassette. The International Expert Group for Cassette Classification SCC*mec* has distinguished eleven types of cassettes. The criterion for belonging to a given subtype is the diversity of the *ccr* gene and the *mec* gene. The structural diversity of the cassettes is so great that many of them cannot be typed at all, which brings many diagnostic difficulties [21,23,24]. Genes, whether *blaZ*, *mecA* or *mecC*, can undergo horizontal transfer. This transfer can occur within the same or different bacterial species. This is a very dangerous phenomenon, leading to the spread of antimicrobial resistance of microorganisms and difficulties in outpatient care [13,19,23].

In view of the above, the aim of this study was to investigate the prevalence and antimicrobial susceptibility profile of staphylococci to a variety of antimicrobial agents (benzylpenicillin, ciprofloxacin, norfloxacin, quinupristin-dalfopristin, amikacin, erythromycin, linezolid, trimethoprim-sulfamethoxazole, tetracycline and chloramphenicol) in the nasopharyngeal microbiota of healthy pregnant women. The presence of common beta-lactam resistance genes (*blaZ*, *mecA* and *mecC*) as well as *ccr* gene types, a class of *mec* cassette and SCC*mec* cassette types was also determined. The highest levels of resistance were found to beta-lactams, commonly used for prophylaxis or treatment during pregnancy, including in daily outpatient practice or short hospital stays. Resistance to a variety of antimicrobials was observed, including benzylpenicillin resistance in *blaZ*-positive isolates and the existence of a very high diversity of SCC*mec* cassette structures in all staphylococci selected from the nasopharyngeal microbiota of pregnant women with a physiological pregnancy course. The present study is, to our knowledge, one of the first studies to evaluate the presence of antimicrobial-resistant staphylococci phenotypes and genotypes in the nasopharyngeal microbiota of pregnant women. This knowledge may be important for the appropriate treatment or prophylaxis of this group of patients, as well as for the fetus.

## 2. Results

### 2.1. Determination of Drug Susceptibility Profile by Disc-Diffusion Method

The study included a total of 68 bacterial isolates of the *Staphylococcaceae* family, with the most prevalent being *S. aureus* 54.4% (37/68) and *S. epidermidis* 27.9% (19/68), followed by *S. hominis* 11.7% (8/68), *S. capitis* 2.9% (2/68), *S. warneri* and *S. saprothyticus* 1.5% (1/68) each.

The disk diffusion method was conducted to determine the susceptibility profile to selected antimicrobials in accordance with the current recommendations of EUCAST [25]. The obtained results are shown in Figure 1, which indicates the highest overall resistance to penicillin and erythromycin. The overall analysis of the obtained data showed that the largest number of *Staphylococcus* spp. isolates showed in vitro susceptibility to aminoglycosides, quinolones and oxazolidinones, with the highest prevalence of all susceptible isolates to quinupristin-dalfopristin.

The percentage analysis of the drug susceptibility profile for each tested staphylococcal species is shown below (Figure 2). It was observed that *S. aureus*, *S. epidermidis* and *S. hominis* showed the highest diversity of drug susceptibility profiles. Both *S. aureus* and *S. epidermidis* showed the highest resistance frequency to beta-lactam antibiotics.

We found 23.5% (16/68) of methicillin-resistant *S. aureus* (MRSA) and 7.4% (5/68) of coagulase-negative *S. epidermidis* (MRCoNSE) and methicillin-resistant coagulase-negative staphylococci other than *S. epidermidis* (nonSE-MRCoNS) each. Penicillin resistance was reported in all staphylococci isolates from the nasopharynx of pregnant women, with the highest frequency of 13.6% (20/68) in methicillin-susceptible *S. aureus* (MSSA) isolates and resistance to erythromycin observed in 7.5% (11/68) of them (Table 1). Resistance to linezolid and norfloxacin was found in only 0.7% (1/68) of MRSA isolates each. Susceptibility, increased exposure of isolates was noted in all isolates to ciprofloxacin in a frequency range of 3.4–12.3%, with the highest percentage noted in MSSA isolates (12.3%, 19/68).

### 2.2. Determination of blaZ, mecA and mecC Gene Prevalence among Tested Staphylococcus spp. Isolates

For each isolate, a PCR by using specific primers for *blaZ*, *mecA* and *mecC* genes was performed.

According to the analysis of the obtained data, the *blaZ* gene was reported in 95.6% (65/68) of the tested isolates, followed by 51.5% (35/68) and 27.9% (19/68) for *S. aureus* and *S. epidermidis*, respectively. The *mecA* gene was detected in 20.6% (14/68) of all tested isolates, in 13.2% (9/68), 4.4% (3/68) and 2.9% (2/68) of *S. aureus*, *S. hominis* and *S. epidermidis* isolates, respectively (Table 2). In addition, the *mecC* gene was detected in only one (1.5%, 1/68) *S. aureus* isolate. The species with the highest frequency of *blaZ* gene presence was *S. epidermidis*, while the *mecA* gene was predominant in *S. hominis* isolates.

### 2.3. Determination of SCCmec Gene Cassette Type

For *mecA*-positive isolates, the presence of either the *ccr* gene type or the *mec* complex type was confirmed by PCR (Table 3). The presence of the *ccr* gene was detected in 11.8% (8/68) of the tested bacteria. Further analysis identified 2.9% (2/68) of *S. hominis* isolates with the *ccrA1* gene (type I *ccr* complex); 5.9% (4/68) with *ccrA4* (type IV *ccr* complex), including two strains of *S. epidermidis* and *S. aureus*; and 4.4% (3/68) with *ccrC* (type V *ccr* complex), including one *S. epidermidis* and two *S. aureus* isolates. One (1.5%, 1/68) of all tested strains was characterised by a double *ccr* allotype (*ccrA4* + *ccrC* detected in *S. aureus* isolate). Further analysis of the obtained results allowed the identification of 5.9% (4/68) of the isolates with a specific *mec* complex class, comprising 4.4% (3/68) *S. hominis* with the presence of the *mecI* gene (class A *mec* complex). The presence of the IS*1272* sequence (class B *mec* complex) was observed in 1.5% (1/68) of *S. epidermidis* strains. For the remaining analysed isolates, the classification was not specified.

The data presented above allowed us to assign the tested isolates to the different types of SCC*mec* cassettes. The results of these tests are shown in Table 4. As a result, only one tested strain could be assigned to SCC*mec* cassette type VI; it was *S. epidermidis*. The remaining strains were classified to UT5v type according to Zong et al. [14]. The final result was that only three tested strains could be assigned to a specific SCC*mec* cassette type, including one *S. epidermidis* and two *S. hominis* isolates.

The frequency of each gene detected in this study in relation to the staphylococcal *mecA*-positive species and the number of isolates obtained from a pregnant patient is shown in Appendix A.

The following diagnostic test reliability parameters were calculated: specificity, sensitivity and positive and negative predictive values for *blaZ* and *mecA* gene detection compared with phenotypic methods performed for beta-lactam resistance testing (Table 5).

The obtained data showed 100% diagnostic test sensitivity and NPV for *mecA* gene detection and over 80% sensitivity and PPV for *blaZ* gene detection in comparison with phenotypic methods used to determine beta-lactam resistance in staphylococci isolated from the respiratory microbiota of pregnant women. Detection of the *blaZ* gene had a higher specificity than *mecA* gene detection (43.75% vs. 18.18%) and a lower NPV value (70.0% vs. 100%).

## 3. Discussion

There are some serious risks associated with giving antibiotics to pregnant women, so there are a number of rules to follow determined by the condition of the patients, the type of examinations and procedures performed and the mode of delivery [1,2,4,5]. Antibiotics with the lowest possible risk of adverse effects should be chosen, which often means choosing older antibiotics with documented safety in pregnancy [1,3]. The beta-lactam group of antibiotics was chosen for this study because the studied isolates were exclusively from women in a physiological state of pregnancy when the choice of treatment is clinically very limited. The negligible risk of pregnancy-related complications, low organ toxicity and easy penetration of the blood–placental barrier make these antibiotics the most commonly used group of antimicrobial drugs during pregnancy [1,3]. Różanska et al. [3] conducted a study to analyse the use of antibiotics prescribed by gynaecologists during pregnancy in one province of Poland. The authors concluded that monotherapy was used in almost 96% of the studied women, and beta-lactams were used in combination with beta-lactamase inhibitors in more than 67% of the patients. These findings are in line with recommendations for antibiotic therapy in pregnant women [1,2]. However, there is still no clear information on which stages of pregnancy are critical in terms of the adverse effects of drugs on the developing foetus [28]. The risks to the foetus from maternal use of antibiotics are also not fully understood. Therefore, the use of antimicrobial drugs in pregnancy is still controversial [1,2,3]. Another limitation to the use of drugs in this group in clinical practice is the problem of drug resistance among microorganisms. Hospital and outpatient strains of *S. epidermidis* are usually characterised by multidrug resistance, including lack of sensitivity to beta-lactam antibiotics [12,13,14].

Our study included 68 staphylococcal isolates from 43 pregnant women. The study included a drug susceptibility profile for fluoroquinolones (ciprofloxacin and norfloxacin), streptogramins (quinupristin-dalfopristin), aminoglycosides (amikacin), macrolides (erythromycin), oxasolidinones (linezolid), tetracyclines (tetracycline) and other groups (trimethoprim-sulfamethoxazole and chloramphenicol). The data were interpreted and analysed according to EUCAST recommendations [25]. Of all staphylococcal isolates identified in this study, 30.9% and 23.5% were methicillin-susceptible and methicillin-resistant *S. aureus* (MSSA and MRSA) isolates, respectively, with the most common resistance to penicillin being 13.6% and 9.5%, respectively. Among coagulase-negative *S. epidermidis* (CoNSE) isolates, the highest frequency of susceptible, increased exposure isolates was 9.5% to ciprofloxacin and 6.1% to penicillin, while among non-coagulase-negative staphylococci (non-CoNS) isolates, it was 4.8% each. No resistance to chinupristin-dalfopristin was found in this study. Interestingly, 38.2%, 20.6% and 1.5% of the isolates phenotypically resistant to beta-lactams carried the *blaZ, mecA* and *mecC* genes, respectively, whereas 25.0% carried none of the tested genes. The results obtained in this study are consistent with data from other authors. In a total of more than 180 *S. epidermidis* strains, Aubry et al. and Ferreira et al. found penicillin resistances of 30% and 72%, respectively [12,20]. It is now reported that the prevalence of *S. epidermidis* strains resistant to penicillin can exceed 90%, as confirmed by Guo et al. (95.5% in clinical strains and 82.1% in colonised strains) [29] and Wang et al. (80–100% of strains analysed from different provinces in China) [30]. The phenomenon of acquiring resistance to antimicrobial agents is very dynamic within this genus. Penicillins, once used as first-line drugs in staphylococcal infections, are now of little use. This phenomenon has led to a situation where strains are resistant to semi-synthetic penicillins, beta-lactamase inhibitors, cephalosporins and even carbapenems [2,3,8,16,28]. The treatment of infections caused by staphylococci presents many difficulties, both in terms of the diagnosis and in choosing the right treatment.

Based on data from other authors and our own, PCR detection of the *blaZ* gene was considered the gold standard for confirming the presence of penicillinase in the tested staphylococci isolates. This choice was also motivated by the Clinical and Laboratory Standards Institute (CLSI) position, which suggests that detection of this gene should be considered in serious cases of infection with *S. aureus* aetiology requiring penicillin therapy [31]. In this study, 95.6% of *blaZ*-positive isolates were confirmed, of which 38.2% were resistant to benzylpenicillin. Less than 11% of all tested isolates lacked the *blaZ* gene and were not resistant to any of the tested antibiotics. These results differ to some extent from those reported by other authors. In the study by Aubry et al. aimed at comparing different phenotypic methods for the detection of penicillinases in *S. epidermidis*, the presence of the *blaZ* gene was detected in a total of 30% of the strains [12]. Similar studies using the disc diffusion method have been reported by Ferreira et al. [20]. Of more than one hundred staphylococcal strains of different tested species, approximately 72% of CNS isolates resistant to benzylpenicillin possessed the *blaZ* gene. Olsen et al. [7] analysed the diversity and frequency of the *blaZ* gene between human and bovine CoNS and *S. aureus*. The authors detected this gene in 143 strains of penicillin-resistant *S. aureus* and CoNS. They found that all penicillin-resistant strains carried *blaZ* and showed a similar organisation of *blaR1* and *blaZ* genes. Similarly, Bagcigil et al. [18] investigated the carriage of the *blaZ* gene in 78 beta-lactamase-positive *S. aureus* isolates from bovine mastitis. They found that all isolates carried the *blaZ* gene. Similar to our study, Soares et al. [13] evaluated the phenotypical and genotypical antimicrobial resistance profile of 100 CoNS species isolated from dairy cow milk. They found that only 6% of staphylococcal isolates were phenotypically susceptible to penicillin and ampicillin, with *S. xylosus* being the most common species, followed by *S. cohnii*, *S. hominis*, *S. capitis* and *S. haemolyticus*.

Other studies have reported the occurrence of the so-called *blaZ* and *mecA* gene correlation phenomena in staphylococci [13,19,22,32]. There is therefore an ongoing debate about the usefulness of detecting both genes simultaneously in *S. aureus*. In our study, we found that both the *blaZ* and *mecA* genes were present in 11.8% of the *S. aureus* isolates and 7.4% of the CoNS isolates, exclusively from *S. epidermidis*, isolated from the respiratory microbiota of healthy women at different stages of pregnancy. Others reported that 4% of *mecA*^+^ *S. xylosus* isolates also carried the *blaZ* gene, which was also detected in 16% of the isolates and all *mecA*^+^ isolates [13]. Authors assumed that the presence of both genes was correlated with phenotypic beta-lactam resistance. Co-occurrence of the *mecA* and *blaZ* genes has also been reported in MRSA, as studied by Okiki et al. in 135 pregnant women with vaginitis [22]. Of the MRSA isolates, 54.5% carried the *mecA* gene, and none of the MSSA or methicillin-resistant or methicillin-susceptible coagulase-negative staphylococci (MRCoNS or MSCoNS) isolates had the *mecA* gene with a prevalence of 30%. The *blaZ* gene was detected in 65% of the staphylococcal isolates, including 72.7% and 22.2% of MRSA and MSSA, respectively, and 15% of MRCoNS; 54.5% of MRSA isolated from women with vaginal discharge and itching had both *mecA* and *blaZ* genes [22].

Infections caused by various species of staphylococci, e.g., *S. aureus* and *S. epidermidis* in particular, can be a problem for infection due to the ever-increasing drug resistance of these bacteria [7,8,16]. In staphylococci, resistance to several antimicrobial drugs may be associated with insensitivity to multiple chemical groups of antibiotics, giving the character of multidrug-resistant strains. Drug resistance, except plasmid-encoded, can also be found on the bacterial chromosome, as in the case of methicillin resistance, conditioned most often by the presence of the *mecA* and/or *mecC* gene [8,14,19,21,22]. This gene occurs within the so-called SCC*mec* gene cassettes [23], classified into individual types I-XI, thanks to the use of different methods for typing SCC*mec* cassettes, usually based on multiplex PCR reaction [23,27,33]. One of the most effective methods, other than whole-genome sequencing, is proposed by Kondo et al. [27] and is still recommended by the International Working Group for SCC*mec* elements (IWG-SCC) [24]. The PCR reaction proposed by Chen et al. [33] allows the determination of the *mec* complex and *ccr* allotype after the reaction in two panels: the first to detect *mecA*, *mecI*, insertion sequence *IS*1272 (class B of the *mec* gene) and *ccrB2* to detect type II and IV of the SCC*mec* cassette and the second to identify *ccrC*, *ccrB1-B4* and insertion sequence *IS*431 (class C2 of the *mec* gene), allowing the determination of cassette types I, III, V and VI. Both methods have their own limitations as they are not yet sufficiently effective in determining the remaining cassette types, including VII, IX, X, XI and the newest types [23]. This study showed that among *mecA*-positive staphylococci isolated from the respiratory microbiota of pregnant women, taken for further analysis of *ccr* complex types and SCC*mec* cassettes, 21.5% of isolates had type IV, 14.3% had type I or V and 7.1% had type IV + V, while 42.9% of strains contained an undetermined *ccr* complex. Of the staphylococcal strains tested for the *mec* gene complex, 21.4% and 7.1% were class A and B, respectively. We also found that only one *S. epidermidis* isolate had the VIth SCC*mec* cassette type, 14.3% and 7.1% of the UT5v and VI types, respectively, according to Zong et al. [14]. The literature data indicate that *S. epidermidis* strains may contain SCC*mec* I-VIII cassettes, while *S. aureus* may contain any of the eleven types [23]. Our data are in agreement with those of Sign-Moodley et al. [21], who demonstrated that not all unknown isolates could be definitively assigned to clearly defined types using the SCC*mec* typing method. Most of these isolates were not conventional types; untypeable elements appeared to be composite SCC*mec* elements consisting of multiple *ccr* gene complexes. This may be due to the ease of transfer of these genes between different bacterial species. It is assumed that only one class of the *mec* complex is involved, although strains with several different *mec* allotypes have been reported in the literature. Many authors have also shown that the majority of untypeable isolates carry more than one *ccr* gene complex [8,21], which is consistent with our data. A significant proportion of MRSA and MRSE have unclassifiable cassettes. This may be the result of an unusual combination of the *mec* complex and *ccr* genes. Such complexes cannot be assigned to any of the known types. Sequencing of the entire element would be required to determine whether these elements represent a single SCC*mec* element with two *ccr* gene complexes or whether the element consists of two separate integrated SCC*mec* elements [21,24].

The classification of strains of *S. epidermidis* and coagulase-negative staphylococci (CoNS) is very important given the ever-increasing drug resistance of these bacteria, including isolates from the respiratory microbiota where staphylococci may change their role from opportunistic to pathogenic [8,16]. The increasing multidrug resistance of methicillin-resistant strains of staphylococci, either *S. aureus* or CoNS, associated with the presence of additional resistance genes to antibiotics is becoming an additional problem for clinicians [8,28]. Increasing resistance to beta-lactams is making antimicrobials, including beta-lactams, an increasingly ineffective therapy for treating staphylococcal infections. This phenomenon significantly limits the therapeutic options available to patients, including pregnant women, for whom there are relatively few treatment options [8,16,28,34]. This in turn may contribute to prolonged antibiotic therapy or discomfort associated with taking antibiotics and/or chemotherapeutics and thus an increased risk of complications. There are also major diagnostic difficulties with coagulase-negative microorganisms as there are no clear diagnostic recommendations for detecting resistance mechanisms in these microorganisms [8,16,23,25,31,34]. The issue of genetically determined resistance and drawing appropriate conclusions especially about CoNS resistance to penicillin is complicated by its production mechanism [16,34]. It occurs in an inducible manner, i.e., only when a beta-lactam antibiotic is present in the bacterial environment [12].

The lack of clear recommendations and reliable methods for the detection of penicillinases in CoNS presents many diagnostic difficulties. The possibility of detecting beta-lactam antibiotic resistance mechanisms by *blaZ* gene detection increases the role of molecular testing in favour of traditional phenotypic methods and their use in daily clinical practice. In the available literature, it is still difficult to find publications that address the issue of the possible consequences of the presence of resistance genes on molecular elements carried by microorganisms belonging to the human microbiota. As shown by the studies carried out in this work and others [12,13,14], CoNS staphylococci, in addition to co-forming the human respiratory microbiota, may represent a potential reservoir of genetic determinants determining phenotypic resistance to antibiotics and/or chemotherapeutics. Furthermore, the overuse and prevalence of antibiotics and their presence in the human body make them optimal carriers of resistance genes.

## 4. Materials and Methods

### 4.1. Bacterial Isolates and Culture

Sixty-eight isolates of *Staphylococcaceae* were used in this study as the clinical material. The bacteria were cultured and identified from nasopharyngeal swabs of 43 adult (19–41 ± 4.6, mean age 29.8 years) healthy women at different stages of pregnancy (2.15%, 5/43 in the first; 3.9%, 9/43 in the second; and 15.9%, 37/43 in the third trimester). The exclusion criteria for patient participation in our study were as follows: the absence of infection at the time of collection, the absence of chronic and acute diseases of the respiratory tract and the absence of antimicrobial use within a 2-year retrospective period. All tested isolates were not an etiological agent of any infection but were derived from nasopharyngeal microbiota as colonizers. All staphylococcal isolates were identified phenotypically based on their properties, e.g., with colony morphology and Gram-staining effect, biochemical features and protein profile detection by the matrix-assisted laser desorption-ionization time of flight mass spectrometry (MALDI-TOF MS) technique. MALDI-TOF mass spectroscopy was performed as described by Bucka-Kolendo et al. [26] on the basis of the protein profile at the Department of Epidemiology and Infectious Diseases, University of Life Sciences, Lublin, Poland. A few colonies of fresh 24 h microbial cultures were aspirated into 1.5 mL Eppendorf tubes containing 150 µL ultrapure water. The tubes were vortexed. Ethanol was added to the suspension at 450 µL, vortexed and centrifuged at 13,000 × *g* for 5 min. Microbial pellets were suspended in 40 µL of 70% formic acid (Fluka Analytical, Munich, Germany). After vortexing, the sample was centrifuged (13,000 × *g*, 2 min) with the addition of 40 µL of acetonitrile (Fluka Analytical, Munich, Germany). For identification, an equal volume of α-cyano-4-hydroxycinnamic acid (HCCA) solution (Bruker Daltonics, Billerica, MA, USA) was added to 1 µL of the protein extract on a MALDI target plate. The MALDI Biotyper 3.1 database—Build (Bruker, Billerica, USA) was used to process the mass spectra obtained for each isolate. The probability of correct identification was expressed in the form of a score index with a range of values: 2.300–3.000, reliable identification of the microorganism to the species level; 2.000–2.299, reliable identification of the microorganism to the genus level and probable identification result to the species level; 1.700–1.999, probable identification result to the genus level; and 0.0–1.699, unreliable identification result.

After that, the bacteria were subcultured on Mueller–Hinton agar medium (MHA, Oxoid, Hampshire, United Kingdom) and incubated for 18 ± 2 h at 35 ± 2 °C under aerobic conditions. The same medium was also used for antimicrobial drug susceptibility testing using the disc diffusion method according to the recommendations of the European Committee on Antimicrobial Susceptibility Testing (EUCAST) [25]. The following American Type Culture Collection (ATCC) reference strains were used: methicillin-resistant *S. aureus* ATCC BAA-1707 (MRSA), *S. aureus* ATCC 29213 (EUCAST quality control) and *S. epidermidis* ATCC 12228.

### 4.2. Disk Diffusion Method

Staphylococcal drug susceptibility was assessed using the disk diffusion method, according to current criteria and EUCAST recommendations [25]. All isolates were analysed for their susceptibility profiles to the selected antimicrobials benzylpenicillin (1 U), ciprofloxacin (5 μg), norfloxacin (10 μg), quinupristin-dalfopristin (15 μg), amikacin (30 μg), erythromycin (15 μg), linezolid (10 μg), trimethoprim-sulfamethoxazole (1.25–23.75 μg), tetracycline (30 μg) and chloramphenicol (30 μg). All isolates were also tested for the production of methicillin resistance mechanisms using a cefoxitin disc (30 μg) [25]. Isolates were defined as cefoxitin-resistant if the diameter of their growth inhibition zone around the antibiotic disc was <22 mm. Such staphylococcal isolates were classified as methicillin-resistant strains [25]. For this purpose, bacterial inoculum of 0.5 McFarland scale in sterile 0.85% NaCl solution from fresh, 24 h culture was prepared. Inoculated plates, inoculated in three directions, with a disc impregnated with a known concentration of antimicrobial compounds placed on top, were pre-incubated at room temperature for 15 min and incubated under the conditions described above. The size of the growth inhibition zone [mm] for each antibiotic/chemotherapeutic disc as a result of the susceptibility testing was used to classify the staphylococcal isolates into the following three categories: susceptible, standard dosing regimen (S); sensitive, increased exposure (I); and resistant (R) [25].

### 4.3. DNA Extraction and PCR Reaction

#### 4.3.1. Specific PCR Reactions

Genomic DNA from each bacterial isolate was extracted by using a lysostaphin-based extraction protocol with the Genomic Mini Kit (A&A Biotechnology, Gdansk, Poland). Detection of the genes *blaZ*, *mecA* and *mecC* was performed by PCR reaction in a total volume of 15 µL, containing 7.5 µL REDTaq^®^ ReadyMix™ PCR Reaction Mix (Merck, Darmstadt, Germany) and 1 µL of bacterial genomic DNA, by using specific primers (Appendix A) and temperature–time procedures as previously published [12,32]. The following controls were included in all amplification reactions: *S. aureus* ATCC 29213 (*blaZ*-positive *S. aureus*), *S. aureus* ATCC BAA-1707 (*mecC*-positive *S. aureus*), *S. aureus* ATCC 25923 (*mecA, blaZ*-negative *S. aureus*), *S. epidermidis* ATCC 12228 (*mecA, mecC*-negative *S. epidermidis*) and negative control (water).

#### 4.3.2. SCC*mec* Cassette Detection and Typing

Further PCR reactions were then performed on all *mecA*-positive (*mecA*^+^) isolates to detect specific sequences and products to determine the type of *ccr* gene complex and the class of *mec* gene complex [27]. Three separate multiplex-PCR (M-PCR) reactions were performed for this purpose (Appendix A). The identification of five types of SCC*mec* elements was performed by M-PCR reactions in a total volume of 25 µL containing 12.5 µL of REDTaq^®^ ReadyMix™ PCR Reaction Mix (Merck, Darmstadt, Germany), 5 µL of DNA, 1 µL of each 10 µM specific primer and the missing volume of molecular-use H_2_O (EURx, Gdansk, Poland).

The types of SCC*mec* cassette were detected in reactions consisting of a pre-denaturation of 94 °C for 5 min and final extension of 72 °C for 2 min and 30 cycles (consisting of the following steps: denaturation of 94 °C for 2 min, annealing of 57–60 °C for 1 min and extension of 72 °C for 2 min), followed by 1.5–2.0% gel electrophoresis. Each gel was stained with SimplySafe dye (EURx, Gdansk, Poland), documented with Quantum VilberLourmat ST4 system and visualised with Quantum-Capt ST4 (Vilber Lourmat, Collégien, France). The following controls were included in all amplification reactions: *S. aureus* ATCC BAA-1707 (SCC*mec*: Type IV-positive *S. aureus*), *S. aureus* ATCC 25923 (*mecA*-negative *S. aureus*), *S. epidermidis* ATCC 12228 (*mecA*-negative *S. epidermidis*) and negative control (water).

### 4.4. Data Analysis

An UpSet plot was constructed to show the antimicrobial resistance pattern of all staphylococcal isolates using online tools (https://asntech.shinyapps.io/intervene/ (accessed 2 February 2023)). An UpSet plot showed the number of shared phenotypes with gene hits that were found in each isolate. In the binary input file, each column represented a set, and each row represented an element. If a name was in the set, it was represented as a 1; otherwise, it was represented as a 0, based on the manufacturer’s instructions.

## 5. Conclusions

Among the tested isolates, the highest resistance was shown to beta-lactam antibiotics, commonly used in outpatient clinics. The presence of the *blaZ* gene determining the synthesis of penicillinases is common in bacteria of the genus *Staphylococcus*. Phenotypic and genotypic tests showed that *blaZ*-positive isolates were resistant to benzylpenicillin. Detection of the *blaZ* gene had a higher specificity than *mecA* detection (44% vs. 18%) and a lower either sensitivity (94% vs. 100%) or NPV value (70% vs. 100%). All staphylococci were found to have a very high diversity of SCC*mec* cassette structures. Knowledge of methicillin-resistant staphylococci in humans, especially the nasopharyngeal microbiota of pregnant women, needs to be further developed. Our findings are of cognitive importance as there is little knowledge of the resistance of bacteria isolated from the respiratory microbiota, including the nasopharyngeal cavity, of women during the physiological course of pregnancy. A better understanding of this phenomenon, given the global increase in resistance, the large number of opportunistic strains currently circulating in the community, and the presence of opportunistic pathogens resistant to antibiotics and other antimicrobials in the maternal respiratory microbiota, may be helpful in preparing treatment or prevention that may be useful in improving pregnancy outcomes.

## Figures and Tables

**Figure 1 ijms-24-07980-f001:**
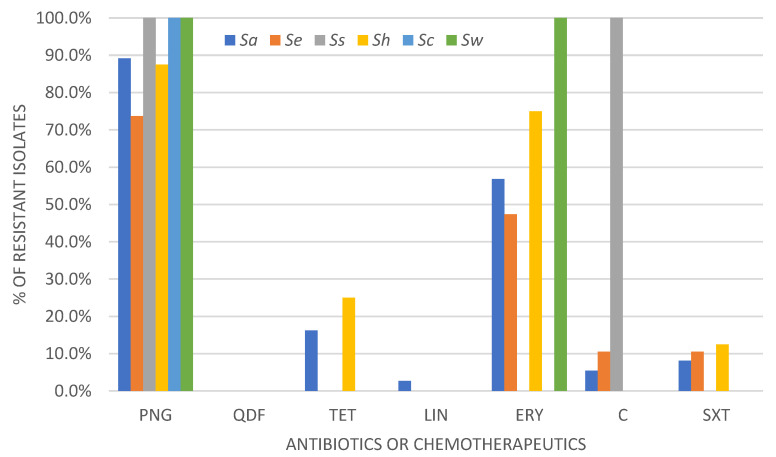
Species-dependent resistance of all tested *Staphylococcus* spp. isolates selected from nasopharyngeal microbiota of pregnant women to individual antibiotics/chemotherapeutics on the basis of the disc-diffusion method. Abbreviations: Sa—*Staphylococcus aureus* (n = 37); Se—*Staphylococcus epidermidis* (n = 19); Ss—*Staphylococcus saprophyticus* (n = 1); Sh—*Staphylococcus hominis* (n = 8); Sc—*Staphylococcus capitis* (n = 2); Sw—*Staphylococcus warneri* (n = 1); PNG—benzylopenicillin (1 U); QDF—quinupristin-dalfopristin (15 µg); TET—tetracycline (30 µg); LIN—linezolid (10 µg); ERY—erythromycin (15 µg); C—chloramphenicol (30 µg); SXT—trimethoprim-sulfamethoxazole (1.25 µg–23.75 µg).

**Figure 2 ijms-24-07980-f002:**
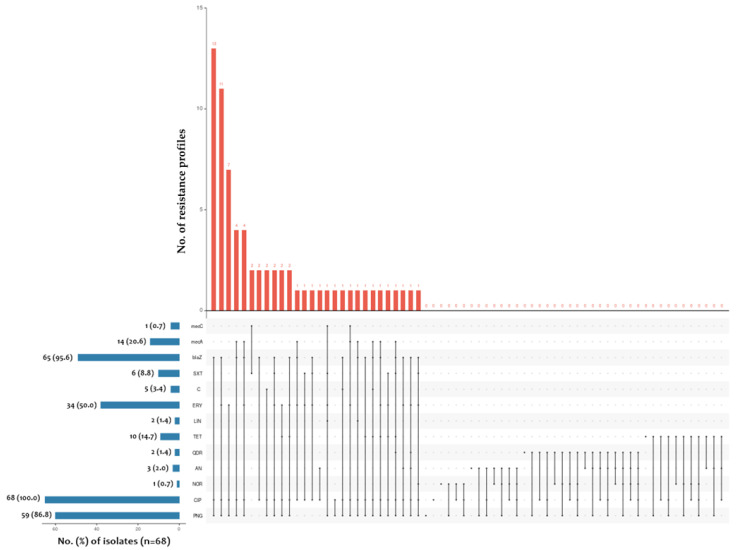
Resistance pattern of all tested *Staphylococcus* spp. isolates selected from the nasopharyngeal microbiota of pregnant women to individual antibiotics. Abbreviations: PNG—benzylopenicillin (1 U); CIP—ciprofloxacin (5 μg); NOR—norfloxacin (10 μg); AN—amikacin (30 μg); QDR—quinupristin-dalfopristin (15 µg); TET—tetracycline (30 µg); LIN—linezolid (10 µg); ERY—erythromycin (15 µg); C—chloramphenicol (30 µg); SXT—trimethoprim-sulfamethoxazole (1.25 µg–23.75 µg); *blaZ*—presence of the *blaZ* gene; *mecA*—presence of the *mecA* gene; *mecC*—presence of the *mecC* gene; left barplot—the total number (%) of resistance to each antimicrobial agent and/or resistance gene represented by studied isolates (n = 68); top barplot—every possible resistance profile and its occurrence in the studied isolates.

**Table 1 ijms-24-07980-t001:** Patterns of resistance to all tested antimicrobials in different phenotypes of staphylococci isolated from pregnant women.

Antimicrobial Drug	% (Number) of Resistant/Susceptible, Increased Exposure Isolates (n = 68)					
	SA (n = 37)		CoNSE (n = 19)		nonSE-CoNS (n = 12)	
	MSSA (n = 21)	MRSA (n = 16)	MRCoNSE (n = 5)	MSCoNSE (n = 14)	nonSE-MRCoNS (n = 5)	nonSE-MSCoNS (n = 7)
PNG	13.6 (20)	9.5 (14)	3.4 (5)	6.1 (9)	2.7 (4)	4.8 (7)
CIP	1.4 (2)/12.3 (19)	3.4 (5)/7.5 (11)	0.0 (0)/3.4 (5)	0.0 (0)/9.5 (14)	0.0 (0)/3.4 (5)	0.0 (0)/4.8 (7)
NOR	0.0 (0)	0.7 (1)	0.0 (0)	0.0 (0)	0.0 (0)	0.0 (0)
AN	0.0 (0)	0.7 (1)	0.0 (0)	1.4 (2)	0.0 (0)	0.0 (0)
QDR	0.0 (0)/0.7 (1)	0.0 (0)/0.7 (1)	0.0 (0)	0.0 (0)	0.0 (0)	0.0 (0)
TET	2.7 (4)/1.4 (2)	1.4 (2)	0.0 (0)	0.0 (0)	1.4 (2)	0.0 (0)
LIN	0.0 (0)	0.7 (1)	0.0 (0)	0.0 (0)	0.0 (0)	0.0 (0)
ERY	7.5 (11)	6.8 (10)	0.7 (1)	5.4 (8)	2.7 (4)	0.0 (0)
C	1.4 (2)	0.0 (0)	1.4 (2)	0.0 (0)	0.0 (0)	0.7 (1)
SXT	0.7 (1)	1.4 (2)	0.0 (0)	1.4 (2)	0.7 (1)	0.0 (0)

Abbreviations: SA—*Staphylococcus aureus*; MSSA—methicillin-susceptible *Staphylococcus aureus*; MRSA—methicillin-resistant *Staphylococcus aureus*; CoNSE—coagulase-negative *Staphylococcus epidermidis*; MRCoNSE—methicillin-resistant *Staphylococcus epidermidis*; MSCoNSE—methicillin-susceptible *Staphylococcus epidermidis*; nonSE-CoNS—coagulase-negative staphylococci other than *Staphylococcus epidermidis*; nonSE-MRCoNS—methicillin-resistant coagulase-negative staphylococci other than *Staphylococcus epidermidis*; nonSE-MSCoNS—methicillin-susceptible coagulase-negative staphylococci other than *Staphylococcus epidermidis*; PNG—benzylopenicillin (1 U); CIP—ciprofloxacin (5 μg); NOR—norfloxacin (10 μg); AN—amikacin (30 μg); QDR—quinupristin-dalfopristin (15 µg); TET—tetracycline (30 µg); LIN—linezolid (10 µg); ERY—erythromycin (15 µg); C—chloramphenicol (30 µg); SXT—trimethoprim-sulfamethoxazole (1.25 µg–23.75 µg).

**Table 2 ijms-24-07980-t002:** The occurrence and relationship of the *blaZ, mecA* and *mecC* gene within staphylococci isolates (n = 68) selected from the nasopharyngeal microbiota of pregnant women and phenotypic resistance to beta-lactams.

Species		No. (%) of Isolates Phenotypically Beta-Lactam Resistant		No. (%) of Isolates with Each Resistance Gene Occurring		
				*blaZ* Gene	*mecA* Gene	*mecC* Gene
SA	*S. aureus*	3 (4.4%)	35 (51.5%)		9 (13.2%)	1 (1.5%)
CoNSE	*S. epidermidis*	5 (7.4%)	19 (27.9%)		2 (2.9%)	0
nonSE-CoNS	*S. hominis*	0	7 (10.3%)		3 (4.4%)	0
	*S. capitis*	0	2 (2.9%)		0	0
	*S. warneri*	0	1 (1.5%)		0	0
	*S. saprophyticus*	0	1 (1.5%)		0	0
Total			65 (95.6%)		14 (20.6%)	1 (1.5%)

Abbreviations: SA—*Staphylococcus aureus*; CoNSE—coagulase-negative *Staphylococcus epidermidis*; nonSE-CoNS—coagulase-negative staphylococci other than *Staphylococcus epidermidis.*

**Table 3 ijms-24-07980-t003:** The *ccr* gene complex types among *mecA*-positive staphylococci.

Type of *ccr* Complex According to [24,26]	% (Number) of Isolates (n = 68)
Type I	2.9 (2)
Type II	0.0 (0)
Type III	0.0 (0)
Type IV	5.9 (4)
Type V	4.4 (3)
Type IV + type V	1.5 (1)
Not defined	8.8 (6)
Total	23.5 (16)

**Table 4 ijms-24-07980-t004:** Types of SCC*mec* gene cassettes.

Strain	Type of *ccr* Complex	Class of *mec* Gene Complex	SCC*mec* Cassette Type According to [27]	Type According to [14]
*Staphylococcus aureus*				
1.	V	-	ND	-
2.	IV	-	ND	-
3.	-	-	ND	-
4.	IV	-	ND	-
5.	-	-	ND	-
6.	-	-	ND	-
7.	-	-	ND	-
8.	V	-	ND	-
9.	I	A	ND	UT5v
*Staphylococcus epidermidis*				
10.	IV	B	VI	VI
11.	IV + V	-	ND	-
*Staphylococcus hominis*				
12.	-	A	ND	-
13.	I	A	ND	UT5v

ND—not determined.

**Table 5 ijms-24-07980-t005:** Diagnostic performance of all methods testing for beta-lactam resistance genes.

	*blaZ*	*mecA*
Specificity	43.75%	18.18%
Sensitivity	94.23%	100.00%
PPV	82.76%	23.73%
NPV	70.00%	100.00%

Abbreviations: PPV—positive prediction value; NPV—negative prediction value.

## Data Availability

Not applicable.

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
