# Peer review of "Staphylococcal Resistance Patterns, *blaZ* and SCC*mec* Cassette Genes in the Nasopharyngeal Microbiota of Pregnant Women"

_ijms, 2023, doi:10.3390/ijms24097980_

Round 1
Reviewer 1 Report
Reviewer report manuscript ID ijms-2284068
Congratulations on the present study!
General comments
The manuscript is well-written and no major comments or English editing are needed. I just found some minor typo errors, please see my minor comments below.
The Materials and Methods section should be improved by the authors for replication purposes to the Readers. Likewise, the Results section showed irregularities in the Figures, which must be amended by the authors. Please see the exact issues in my minor comments below.
Lastly, the Discussion section is well-written, but it is lacking the acknowledgment of the several shortcomings of the present study, which is necessary for non-familiar Readers.
Minor comments
Introduction
Lines 98-100- Please add the methodologies applied for the evaluation of the resistance in the study.
Materials and Methods
Line 111- Please clarify for the Readers about the equipment model and database used in the MALDI-TOF MS analysis.
Lines 127-129- Please clarify if the sentence “Isolates were defined as cefoxitin-resistant if the diameter of their growth inhibition around the antibiotic disc was < 22 mm.” is based on EUCAST recommendations or other references, if so, please cite it.
Section 2.3.1- The primers’ amplicons are very small and no report was done on the positive and negative controls for PCR assays. Please clarify this matter to the Readers.
Section 2.3.2- Please clarify again about the controls used in the SCCmec cassette detection and typing.
Section 2.4- Please elaborate on the data analysis for Readers non-familiar with this evaluation.
Results
Section 3.1- Figures 1 and 2 are wrongly cited in the text, first, the authors should cite Figure 1 and then Figure 2. Also, the second figure (Figure 1) has the second plot (the one with blue bars) with wrong percentage values. Please rectify these mistakes.
Figure 3- Please improve the resolution of the picture for the Readers.
Discussion
Line 359- Please put ¨S. aureus¨ in italics.
Page 12- The study possesses several shortcomings, and no recognition was done b the authors. The study is an excellent work and deserves to be published. However, the authors must clarify this theme to non-familiar Readers. This is not a recommendation.
Congratulations again to the authors for the present study.

Author Response
Thank You very much for Your participation and your very valuable comments on our manuscript. All comments and corrections have been included in the attached file.

Reviewer 2 Report
Title
I strongly suggest changing the title. Isolates were not obtained from the entire respiratory tract, but from the noses of patients, so it is difficult to speak of a representation of the entire respiratory tract bacteribiome.
Materials and Methods
Why were the strains maintained on MH medium? Bacteria belonging to the genus Staphylococcus do not have special nutritional requirements. And the passage of bacteria through porous substrates affects their phenotype in a negative way.
What database was used for MALDI-TOF MS for identification?
So all the strains obtained have been confirmed with MALDI?
Since several isolates were obtained from individual patients - at least a table should be added in the supplement, which shows this fact.
Disk diffusion method
Why wasn't the D-test done? The occurrence of resistance to clindamycin is probably a very important issue, especially since clindamycin is a frequently used antibiotic, as the authors mention in the introduction. And all of us have been exposed to clindamycin at some point in our lives.
Disk diffusion method
Why was the E-test used for only one substance? it is very confusing. All isolates that show patterns of resistance in the disc diffusion test should have specific MIC values for the individual substances. Why is the MIC value not specified? What, however, is the basis in this type of research.
Table 2
What primers were used for multiplex PCR? Table 2 and Table 1 could easily be included in the supplement.
Figure 3
It is unnecessary.
Patients
Why is the topic of possible sources of resistance in patients not addressed? A history of antibiotic therapy in the past should be made and additionally, e.g. in the form of a table attached as a supplement. The quality of the origin of resistant strains in relation to the patients themselves, not in general, and comparing this with literature data should be a key element of the discussion.
Author Response
Thank you very much for your participation and your very valuable comments on our manuscript. All comments and corrections have been included in the attached file.

Reviewer 3 Report
The manuscript is of scope and is well written. However, there are many things that authors need to work on following sections.
Abstract:
The abstract needs to be completed. The abstract should begin with a brief but precise statement of the problem or issue, followed by a description of the research method and design, the major findings, and the conclusions reached.
Introduction:
Add some more background information so that the readers can understand, and finally, summarize your main points.
Overall, the authors need to restructure the manuscript. Besides, I felt there are fewer supporting references supporting the statement.
The manuscript is of scope and is well written. However, there are many things that authors need to work on following sections.
Abstract:
The abstract needs to be completed. The abstract should begin with a brief but precise statement of the problem or issue, followed by a description of the research method and design, the major findings, and the conclusions reached.
Introduction:
Add some more background information so that the readers can understand, and finally, summarize your main points.
Overall, the authors need to restructure the manuscript. Besides, I felt there are fewer supporting references supporting the statement.
Author Response

(The authors gave the same response as above.)

Round 2
Reviewer 2 Report
none